# The acoustical behavior of a bass guitar bridge with no saddles

Jonathan A. Kemp [1,2] *

**1** Music Centre, University of St Andrews, St Andrews, Fife, United Kingdom, **2** SUPA, School of Physics & Astronomy, University of St Andrews, St Andrews, Fife, United Kingdom

* jk50@st-andrews.ac.uk

**Data Availability Statement:** The data underlying the results presented in the study are available from https://osf.io/tbzdc/ which is DOI 10.17605/OSF.IO/TBZDC.

**Funding:** The authors received no specific funding for this work.

## Abstract

The acoustics of a bass guitar bridge without saddles was tested experimentally and the results contextualised. Conclusions were obtained demonstrating that the bridge without saddles (where knot around the ball end of the string forms part of the sounding length) produced no measurable reduction in sustain and may increase the sustain for lower pitched strings, in comparison to a conventional bridge featuring saddles. The bridge without saddles showed a reduction in string inharmonicity, and produced a splitting of the frequency peaks associated within the resonances of the string. This peak splitting is explained as being due to differences in the frequency of vibrations parallel to and perpendicular to the body. Since the loop of core wire strongly resists vibration perpendicular to the body but vibrates freely as part of the sounding length for vibration parallel to the body, the relative length of the loop of core wire with respect to the sounding length of the string determines the fractional difference in frequency. The perceptual quality of the sound is similar to the beating due to multiple strings per note (as in piano) and to electronic chorus effects.

## Introduction

Conventional bass guitar bridges support the string tension through the ball end being pulled through a constriction in the form of metal ferrules inserted into the body of the instrument as in the original design of Fender's precision bass of 1951 (as seen in the patent of the design [1]) or holes in a metal bridge plate as in the 1957 redesign of the instrument bearing the same name (as visible in Fender's later pickup design patent [2]). In either case, each string then bends over a saddle. The bigger the break angle at the saddle, the bigger the force that keeps the string attached to the saddle (rather than bleeding vibrational energy into the portion of string behind the saddle). Large break angles also result in a bigger force keeping the saddle from moving around on the bridge plate, thus maintaining good sustain and stable string spacing. Bass guitar strings are relatively stiff under bending. A large break angle results in a permanent bend or kink in the string and in extreme cases this could lead to layers of windings rubbing against each other during string vibration and/or the string bulging over the saddle rather than following its curvature. This could cause problems with tone and sustain. It could also lead to a longer time taken to achieve stable tuning after installing the string (as the permanent bend in the string takes time to develop). Most bass guitar bridges therefore are based on an acceptable compromise that has, by and large, stood the test of time.

**Competing interests:** The authors have declared that no competing interests exist.

The "Ray Ross Saddle-Less Bass Bridge" is a unique bridge design (patented as US patent number 10,388,261 B1 by Ray Aaron Ross [3]) that uses individual "tone pins", one per string, so that the vibrating length of the string extends all the way to the ball end. The Ray Ross bridge eliminates the need for the bend or kink as the vibration carries on in a straight line all the way to the ball end. Clearly this involves the knot (by which the core is tied to the ball end) forming part of the sounding length of the string. This is not how string manufacturers designed their strings to be used. It is therefore appropriate to give detailed scientific data and analysis to show what measurable changes can be determined in the resulting vibration of the string in comparison to conventional bridge designs.

In this study the acoustical properties of the Ray Ross bridge and a conventional bridge design are compared. Comparisons are drawn based on sustain (time taken for resonances to reduce by 60 dB after plucking the string), inharmonicity (the extent to which resonances tend to go sharp of a harmonic series due to string stiffness), and splitting of frequency peaks (which result from differences between horizontal and vertical modes of vibration). The examples of a typical grand piano note (where two or more strings are deliberated tuned to slightly differing frequencies to create complex beating and to control decay rates in the aftersound [4]), and an electric bass guitar with conventional bridge playing through an electronic chorus effect, were shown to give context to the discussion of peak splitting.

Since theoretical considerations predict that the inharmonicity is reduced to the greatest extent for the thinnest ($G_2$) string on the bass guitar, the experiments involve sensitive measurements using an optical pickup of the correct size for that string to validate the theory. In order to ensure generality of results when considering sustain in realistic musical context, all four strings from a standard set of strings from a different manufacturer were measured in the later experiments with magnetic pickups installed.

## Description of the bridge and string design

An example of a Sadowsky Blue (steel hexagonal core with stainless steel windings) $G_2$ string with a nominal diameter of 0.045 inches as seen in Fig 1. The loop of core wire can be seen. Also, the effect of the twist of core wire that ties the core onto the ball end is visible in the

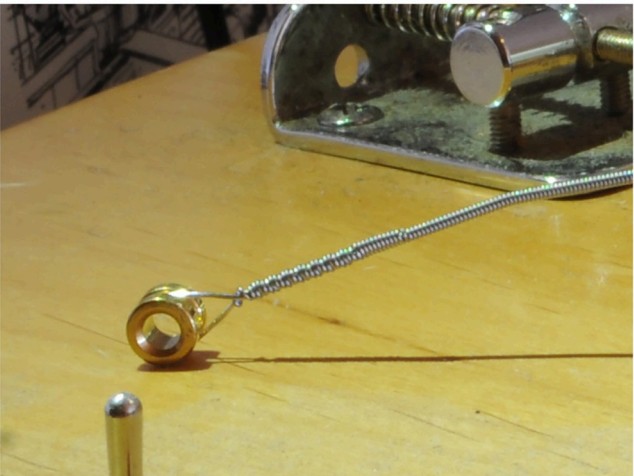

**Fig 1. String.** The bass guitar string, nominally 0.045 inches diameter and designed for $G_2$ pitch, seen uninstalled from the instrument.

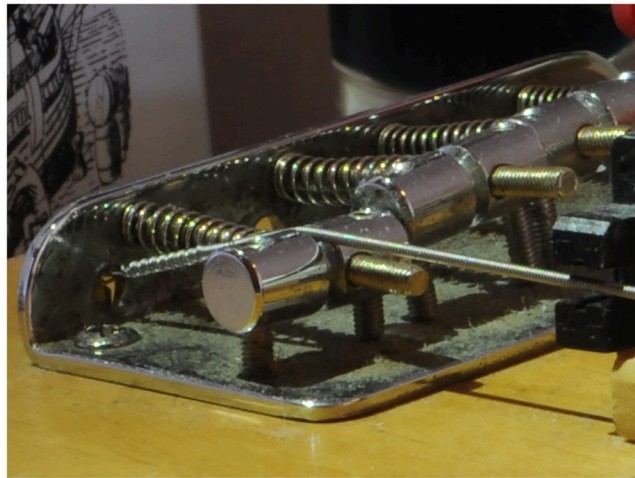

**Fig 2. Stock Yamaha bass guitar bridge.** The loop and knot are located entirely to the left of the saddle and the string tension bends the string over the saddle, forming a kink at the start of the sounding length.

profile of the overlying windings that start at the junction of the knot. Such a string can be seen installed on the stock Yamaha bridge (that is based on Leo Fender's original design) in Fig 2. The kink in the string resulting from installing, tuning, and uninstalling the string from the stock Yamaha bridge may be seen towards the right hand side of Fig 1. A nominally identical string on the Ray Ross bridge is seen in Fig 3 (with the height of the tone pins adjusted to give the same string position as for the stock bridge).

## Theory for the effect of the knot

It is helpful to establish and validate a theoretical method of predicting the effect of including the knot within the sounding length for given string designs. The effect of a raised mass per

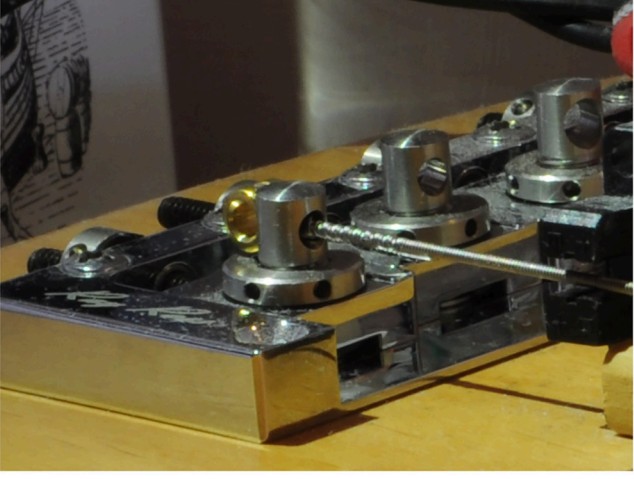

**Fig 3. Ray Ross bass guitar bridge.** The string is installed such that both a portion of the loop of core wire (that passes vertically over the ball end), and the knot that ties the core onto the ball end, form part of the sounding length of the string.

unit length near the end of the sounding length of a string has been shown to be a reduction in inharmonicity [5–9]. This previous work must be extended to simulate the knot which is formed by twisting the wires together at a shallow angle, approximately doubling the mass per unit length of the core (rather than the tight wrapping used when adding windings that would approximately treble the diameter). The most straightforward way of doing this is to simulate the effect of the core knot (where two thicknesses of core wire are present instead of one) by adding a fictitious section of cylindrical winding of diameter $d_f$ within the calculation such that it contributes an equal mass per unit length to that of main core wire in the vicinity of the knot. Equation (2) from [8] gives the ratio of mass per unit length in the constructed string to the mass per unit length of the core, and cancelling terms to simplify gives:

$$\tau = 1 + \frac{\pi^2 \rho_w}{4\gamma_{core}d_1^2\rho_{core}}\left(d_M^2 - d_1^2\right),\tag{1}$$

where $d_1$ is the diameter of the core at its widest point and $d_2$ is the width across the core plus the first section of winding and $d_M$ is the width across the full constructed string with a core plus $M-1$ layers of winding etc., and $\gamma_{core} = 3\sqrt{3}/2$ is the ratio of cross sectional area to radius squared assuming a hexagonal core wire (whereas $\gamma_{core} = \pi$ would be used if simulating a cylindrical core wire). If we set the mass per unit length of the fictitious winding equal to the mass per unit length of the core (setting $M = 2$ and $\tau = 2$ to temporarily consider the knot only), we get the relationship:

$$d_2(\tau = 2) = d_1\left(\frac{4\gamma_{core}}{\pi^2} + 1\right)^{1/2}.\tag{2}$$

The diameter of the "fictitious winding" (which is the diameter of spiral winding around a straight core that gives the doubling of linear density expected of a knotted core) is then:

$$d_f = (d_2 - d_1)/2 = \frac{d_1}{2}\left(\left(\frac{4\gamma_{core}}{\pi^2} + 1\right)^{1/2} - 1\right).\tag{3}$$

The hexagonal cross-section core for the G strings used in these experiments has an approximate minimum diameter (between the flats of the hexagon) of $d_{spec} = 0.017$ inches. This equates to a distance between the points of the hexagon for the core of $d_1 = (2/\sqrt{3})d_{spec} \approx 0.020$ inches and therefore, using Eq 3, a "fictitious winding" of approximate diameter $d_f = 0.004$ inches over the core will model the acoustical effect of the knot within the section of string where the core is wrapped around itself near the ball end.

Assuming the string is oscillating parallel to the body of the instrument, the string may be modelled as consisting of three sections. The first section (labelled $j = 1$ in Fig 4) consists of the loop of core between the ball end and the junction of the loop and has a mass per unit length approximately double that of a single length of core. This loop section has an axial length of $a_1$ and thus $\tau(j = 1) = 2$. Taking the twist of the knot as having a length $a_2$, the $j = 2$ section has both the fictitious winding (to represent the mass in the twist of the knot) of diameter $d_f$, and the real winding of diameter $d_w \approx 0.015$ inches wrapped over that. Finally, the main length of string ($j = 3$) has only a winding of $d_w = 0.015$ inches over the single core. A schematic of this model is shown in Fig 4 and the resulting outer diameters given in Table 1. The theoretical outside diameter for the main section of the constructed string is theoretically 0.050 but in reality this is closer to 0.045. This slight discrepancy is mainly due to the winding becoming slightly oval during manufacture. A string length of 855 + 19 = 874 mm (a little longer than the nominal scale length of 34 inches) was measured for the string.

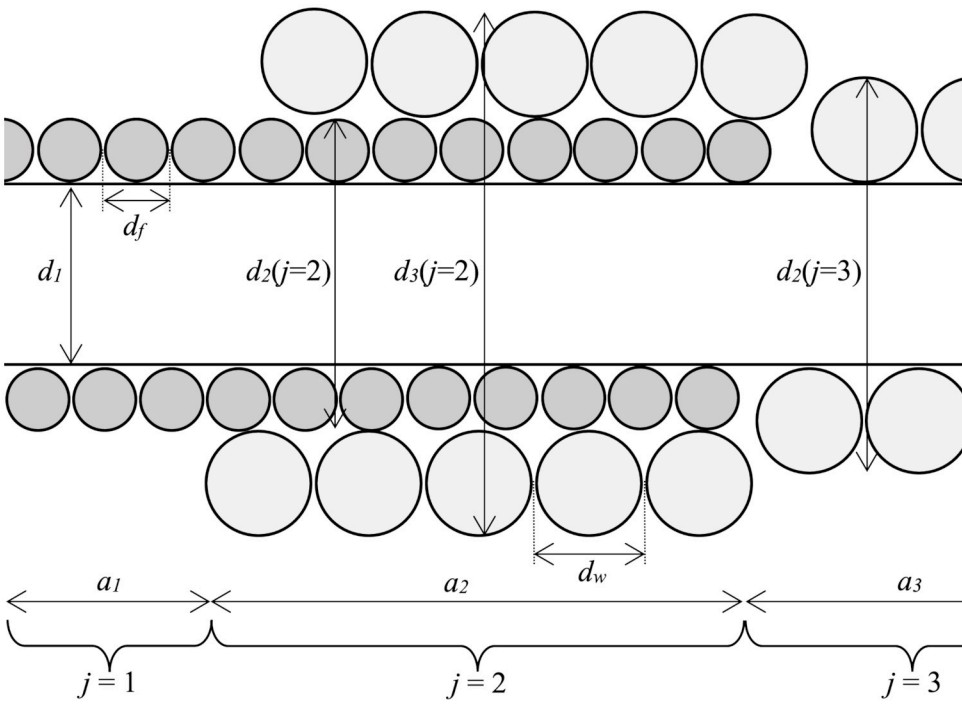

**Fig 4. Schematic of a wound string approximating the effect of a loop and knot.** The fictitious winding of diameter $d_f$ simulates the doubled core in the loop ($j = 1$ section) and in the knot under the winding ($j = 2$ section). The $j = 3$ section constitutes the main sounding length of string with winding of diameter $d_w$.

When the string oscillates perpendicular to the plane of the body then the fixed end is at the junction of the core wire (with the loop of core nearest the ball end omitted from the sounding length due to its very high resistance to displacements perpendicular to the body). The resulting string model is given in Table 2. It should be noted that the sounding length is shorter (855 +16 = 871 mm) for the vibration perpendicular to the body, hence the somewhat higher frequency of vibration for the modes.

The mode frequencies were then calculated using the perturbation theory method by application of the following equations [8]:

$$f_p^{(pert)} = f_p(1 + s)^{-\frac{1}{2}}, \qquad (4)$$

**Table 1. String dimensions relevant for vibrations parallel to the plane of the body.** Approximate section lengths, diameters and mass ratio, $\tau$, for a bass guitar string approximating the specification for Sadowsky Blue nominal 0.045 inch gauge G string when installed on Ray Ross bridge such that the loop section ($j = 1$ here) forms part of the sounding length when considering vibration parallel to the plane of the body.

|  | $j = 1$ | $j = 2$ | $j = 3$ |
|---|---|---|---|
| $a_j$ (mm) | 3 | 16 | 855 |
| $d_1(j)$ (inch) | $d_1 = 0.020$ | $d_1 = 0.020$ | $d_1 = 0.020$ |
| $d_2(j)$ (inch) | $d_1 + 2d_f = 0.028$ | $d_1 + 2d_f = 0.028$ | $d_1 + 2d_w = 0.050$ |
| $d_3(j)$ (inch) |  | $d_2 + 2d_w = 0.058$ |  |
| $\tau(j)$ | 2.00 | 8.38 | 6.12 |

**Table 2. String dimensions relevant for vibrations perpendicular to the plane of the body.** Approximate section lengths, diameters and mass ratio, $\tau$, for a bass guitar string approximating the specification for Sadowsky Blue nominal 0.045 inch gauge G string when installed on Ray Ross bridge (such that the loop section does not form part of the sounding length) when considering vibrations perpendicular to the plane of the body.

| | $j = 1$ | $j = 2$ |
|---|---|---|
| $a_j$ (mm) | 16 | 855 |
| $d_1(j)$ (inch) | $d_1 = 0.020$ | $d_1 = 0.020$ |
| $d_2(j)$ (inch) | $d_1 + 2d_f = 0.028$ | $d_1 + 2d_w = 0.050$ |
| $d_3(j)$ (inch) | $d_2 + 2d_w = 0.058$ | |
| $\tau(j)$ | 8.38 | 6.12 |

which gives the frequency of the $p$th mode, $f_p^{(pert)}$ in terms of the (unperturbed) mode frequencies of a stiff string:

$$f_p = pf_0(1 + Bp^2)^{\frac{1}{2}}, \quad p = 1, 2, 3, \ldots \tag{5}$$

with

$$B = \frac{\pi^2 ES\kappa^2}{4L^4 \tau\mu_{core} f_0^2}, \tag{6}$$

where $E$ is the Young's modulus of the (steel) core, $S$ is the cross-sectional area of the core, $\kappa = \sqrt{5/6}(d_1/4)$ is the radius of gyration of the hexagonal cross-section core, $L$ is the sounding length of the string, $\tau\mu_{core}$ is the mass per unit length of the main sounding length of the string, and

$$f_0 = \frac{1}{2L}\sqrt{\frac{T}{\tau\mu_{core}}}, \tag{7}$$

is the ideal fundamental (if both stiffness and perturbations had been ignored). Finally, the perturbation in Eq 4 requires the factor [8]:

$$s = \sum_{j=1}^{J-1} \left(\frac{\tau(j) - \tau(J)}{\tau(J)}\right)\left(\frac{x_j - x_{j-1}}{L} - \frac{\sin\left(\frac{2\pi p x_j}{L}\right) - \sin\left(\frac{2\pi p x_{j-1}}{L}\right)}{2\pi p}\right), \tag{8}$$

where $J = 3$ is the number of sections in the case of vibrations parallel to the body. The vector $x_j$ consists of the $x$ coordinates of the changes in density with $x_0 = 0$ and

$$x_j = \sum_{i=1}^{j} a_i. \tag{9}$$

Note that Eq 8 dictates that the inharmonicity reduction due to raised mass near the end of the sounding length is most effective when $2\pi p x/L = \pi/2$. Given that the wavelength of the $p$th mode is approximately $\lambda_p \approx 2L/p$, this implies the inharmonicity is reduced to the greatest extent around modes with $\lambda_p \approx 8x$, so where the knot length is around an eighth of the wavelength of the mode under consideration. For a knot of total length 19 mm this maximum reduction in inharmonicity therefore should occur around the 11th or 12th modes for vibration parallel to the body and around the 14th mode for vibration perpendicular to the body assuming a total string length of approximately 874 mm.

Theoretically, a string with no winding would see a doubling of $\tau$ in the knot section, with the strength of inharmonicity reduction being controlled by the factor $(\tau(j) - \tau(J))/\tau(J) = 1$ in Eq 8. It can be shown that in the limiting case of the overall width of the string being much greater than the core width ($d_M \gg d_1$) we obtain $(\tau(j) - \tau(J))/\tau(J) \approx 4d_f/d_M$ at the knot which is much smaller than 1. In practical terms, the $G_2$ string on the bass guitar sees the factor $(\tau(j) - \tau(J))/\tau(J) = 0.37$ to 2 significant figures as deduced from data in Tables 1 and 2. The reduction in inharmonicity is expected to be smaller for the thicker strings as the core size does not usually increase linearly with the overall width of the string (in order to prevent excess stiffness and prevent reduced engineering strain in the core). An example of a $B_0$ string (usually the thickest string on five string electric bass guitar) is seen in recent work [8] as having $\tau = 16.0$ in the main section of string, and adding a knot (modelled using Eq 3) within the sounding length of such as string would give $\tau = 19.5$ there, hence giving the factor $(\tau(j) - \tau(J))/\tau(J) = 0.22$ to 2 significant figures. This means that strength of the factor acting to reduce the inharmonicity for such a $B_0$ string is almost of halved in comparison to that for the $G_2$ string discussed above.

## Experimental comparison of bridge designs for G string using optical pickups

Magnetic drag is known to modify the resonant frequency of modes of vibration perpendicular and transverse to instrument bodies [10]. In order to remove this effect from the initial experiments, a solid body Yamaha BB 350 Natural bass guitar was used with the magnetic pickups removed. The string vibrations were sensed at a position 34 mm from the start of the sounding length using a TCST2103 Transmissive Optical Sensor with Phototransistor Output made by Vishay Semiconductors (which has a 3.1 mm gap between emitter and detector and 1 mm aperture width) in a simple circuit consisting of two resistors, an electrolytic capacitor for AC coupling and 9V battery. A jack input from the pickup circuit was then recorded using a RME Fireface 400 audio interface channel with a gain of +10 dB. Strings were plucked with a plectrum approximately 10 mm from the nut to ensure that all harmonics of interest were excited clearly. The bass was clamped to a worktop at the body and nut and one Sadowsky Blue (steel hexagonal core with stainless steel windings) $G_2$ string with a nominal diameter of 0.045 inches was installed and tuned to pitch.

### Peak detection for G string modes

The results were analysed to measure the frequency, amplitude and Q factor of the resonances within the sound. These resonances are frequently called harmonics, although strictly speaking the resonances go slightly sharp of a true harmonic series due to inharmonicity caused mainly by the bending stiffness of the core of the string [8]. Peaks were searched for, starting with the fundamental frequency of the vibration and then going up the series, one mode at a time, increasing the frequencies bounds for searching to lie above the previous peak detection by 0.75 to 1.25 times the frequency difference between peaks. Detection was performed using the findpeaks function built in to the MATLAB programming environment to find the location and half-height of peaks in the power spectrum (where the power spectrum was taken to be the absolute value of the square of the FFT of the signal). The minimum peak height ('MinPeakHeight' input argument in the find peaks function) was set to 50 times the median of the absolute value of the power spectrum in the search range in order to exclude peaks that were not significant compared to the noise floor. Minimum peak prominence was also set to exclude narrow peaks that are outliers from the noise floor (using a 'MinPeakProminence' of 75).

### Sustain for G string modes

Denoting the frequency of the peak for mode number $p$ as being $f_p$, and defining the half power peak width, $\Delta f$, as the range (in Hertz) of frequency bins in the discrete Fourier transform that were continuously within 3 dB of each peak, the formula

$$Q = \frac{f_p}{\Delta f} \tag{10}$$

was then used to determine the Q factor of each peak.

It may be noted that the amplitude of a damped harmonic oscillator dies off with a factor $\exp(-\alpha t)$ where $t$ is the time in seconds and $\alpha = 2\pi f_p/(2Q)$. This can be used to show that the time taken for 60 dB decay is given by:

$$T_{60} = \frac{6 \ln(10)}{2\pi \Delta f}. \tag{11}$$

The results for deducing the $T_{60}$ using detected peak width is shown in Fig 5. Also shown is a calculation of the reverberation time in third octave bands directly from the audio file using

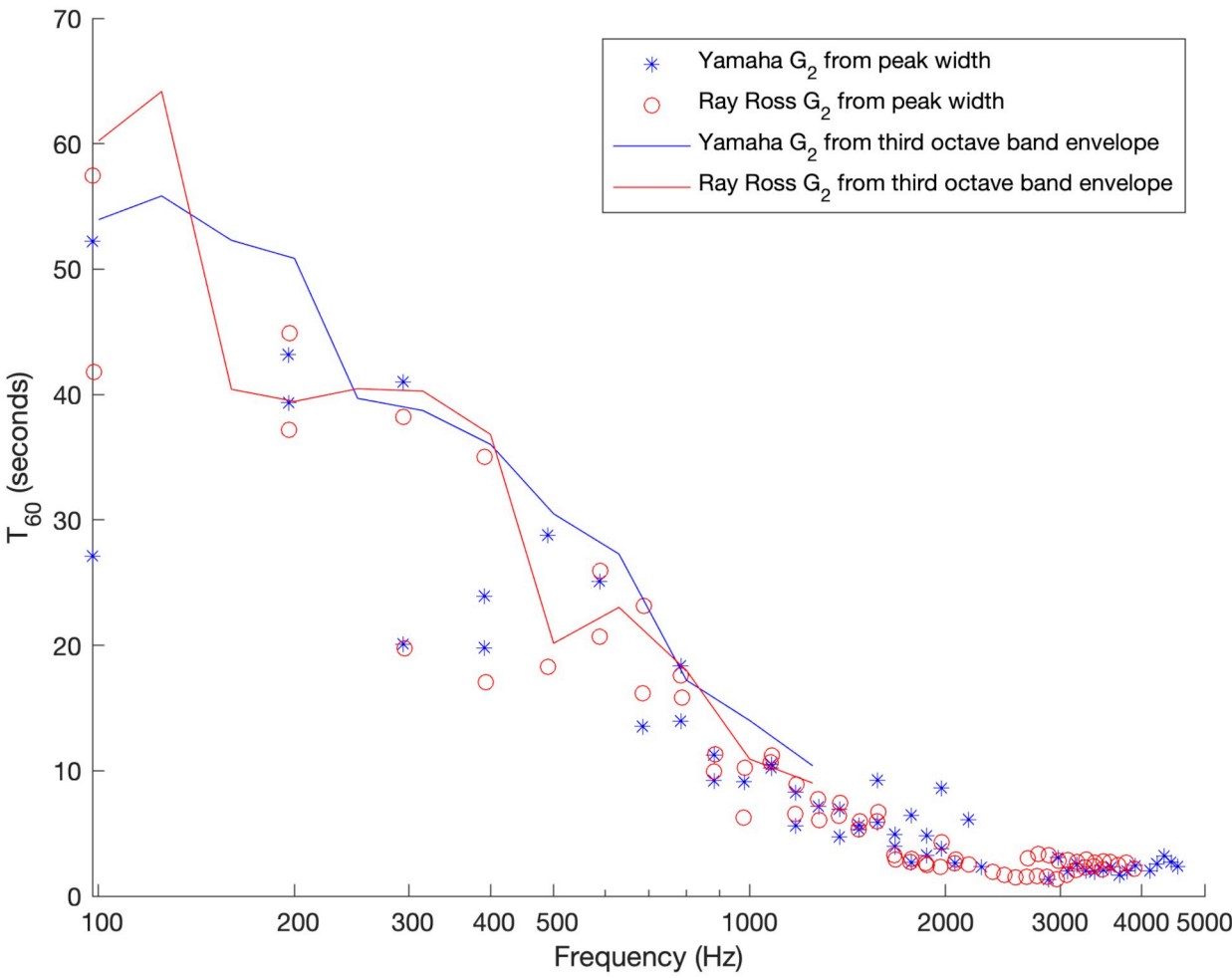

**Fig 5. Sustain for G string modes.** Experimentally determined $T_{60}$ time for the stock Yamaha bridge versus the Ray Ross bridge for the note $G_2$, both measured with optical pickups. Reverberation times based on detecting the width of individual peaks, and based on the time taken for the overall envelope of sound to decay within third octave bands are given.

the t60_impulse function with 't30' input argument from the python-acoustics module (acoustics version 0.2.4). This detects the time taken for the bandpass filtered signal to decay from 5 dB down to 35 dB below the peak level and multiplying the number by two to get the time expected for a 60 dB drop in level. It should be noted that the $T_{60}$ time, which is a measure of how long particular resonances of the instrument sustain for, is very similar for the two bridges, demonstrating that there is no clear difference in sustain of the G string motion due to the inclusion of the knot at the ball end within the sounding length of the $G_2$ string in the Ray Ross bridge versus the inclusion of the bend at the saddle for the Yamaha stock bridge.

It may be noted that for many values of $p$ there were two separate peaks detected at very similar frequencies within the frequency bounds being searched over. This is visible through pairs of data points being in almost the same position horizontally in Fig 5 (for both bridge designs). This occurs because the modes of vibration of the string parallel to and perpendicular to the body of the instrument are occurring at slightly different frequencies for both bridges and the $T_{60}$ times deduced from the widths of both peaks are shown. This "peak splitting" will be discussed in detail in the following sections.

The 25th harmonic for the Yamaha bridge data has failed detections as the optical sensor was mounted close to a node for the 25th harmonic for both horizontal and vertical string motion polarisations in Fig 5. The node positions for the Ray Ross bridge differ for horizontal and vertical polarisations and will be investigated below.

## Inharmonicity for G string modes

Inharmonicity, the tendency of resonances to go progressively sharp of the harmonic series with increasing mode number, can be measured as:

$$\mathbb{I}(\text{cents}) = 1200 \log_2\left(\frac{f_p}{pf_1}\right), \tag{12}$$

where $f_p$ is again the measured resonant frequency of the $p$th mode and $f_1$ is the resonant frequency of the lowest resonant frequency of the string. As set out above, when the core of a guitar or bass string is twisted into a knot in order to attach it to the ball end at the end of the string the result is, approximately, a doubling of the mass per unit length of the core in the first centimetre or two of the string and this will tend to reduce the inharmonicity in comparison to a uniform string. In order to verify this prediction experimentally, the inharmonicity was plotted as shown in Fig 6. Where two peaks were detected in the $f_1$ range, the average of the two frequencies was used for $f_1$ in Eq 12.

The inharmonicity is clearly reduced due to the extra mass near the end of the sounding length in the Ray Ross bridge design. Significant differences in inharmonicity are observed between the pairs of peaks at the same mode number for the Ray Ross bridge due to the modes perpendicular to and parallel to the body having much greater differences for this design of bridge in comparison to the conventional Yamaha bridge design. An example of such a double peak, visible in the FFT data, is shown in the subsequent section. The pairs of peaks are separated by between 5.6 to 9.0 cents for the open G string on the Ray Ross bridge.

## Peak splitting for the G string modes

The spectrum was produced by plotting the absolute value of the discrete Fourier transform, expressed in dB, of the waveform recorded from the optical pickup signal for a plucked note. Double peaks were clearly visible for all resonances in the spectrum in the case of the Ray Ross bridge. For the Yamaha stock bridge, doubled peaks were present but the differences in frequency were much smaller. Fig 7 shows the spectrum zoomed in to around the 4th resonance

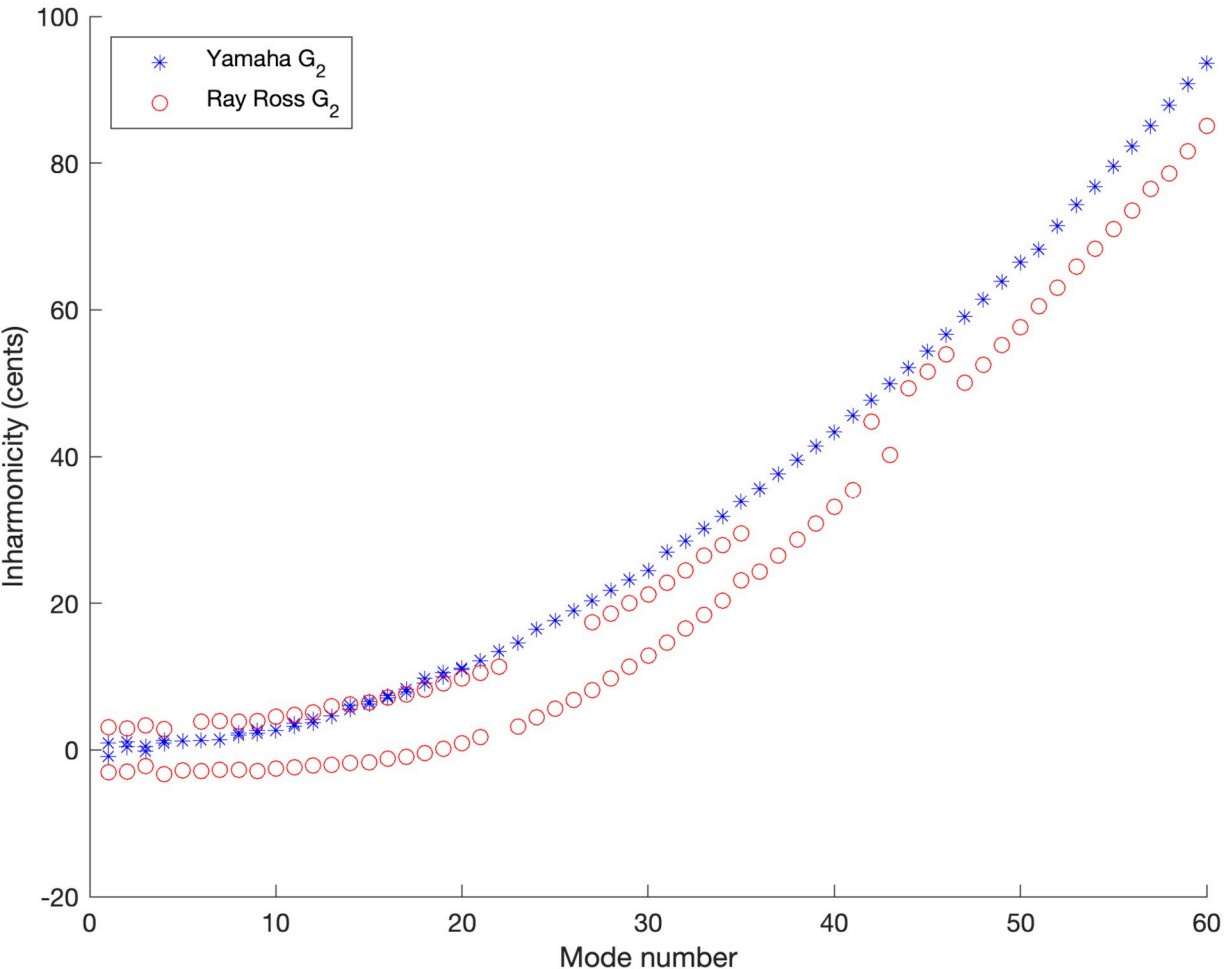

**Fig 6. Inharmonicity.** Experimentally determined inharmonicity for the stock Yamaha bridge versus the Ray Ross bridge for the note $G_2$ as measured with optical pickups.

is order to illustrate the difference clearly. The tuning difference between the two peaks within the Yamaha stock bridge recording was 0.4 cents to one decimal place. In the case of the Ray Ross bridge this difference was 6.1 cents to one decimal place.

As mentioned previously, peak splitting is caused by differences in the modes of vibration for motion perpendicular to and parallel to the plane of the body. For the Yamaha stock bridge (in the absence of magnetic pickups), the very slight difference in frequency between the two peaks is equivalent to what would be expected for two otherwise identical strings with a length difference of around 0.2mm (given the linear relationship between frequency and standing wave length). This is less than the thickness of the string and consistent with slight differences in flexing of the string in the plane of, and perpendicular to the plane of the saddle which the base of the string is in contact with.

In the case of the Ray Ross bridge, the peak splitting can be readily understood in terms of the knot which ties the core onto the ball end. The design of the "tone pins" ensures that the notch in the ball end (and therefore the loop of core wire that wraps over the ball end) sits perpendicular to the body of the instrument. Vibration parallel to the body of the instrument can occur over the entire string length with a fixed end where the string leaves contact with the ball

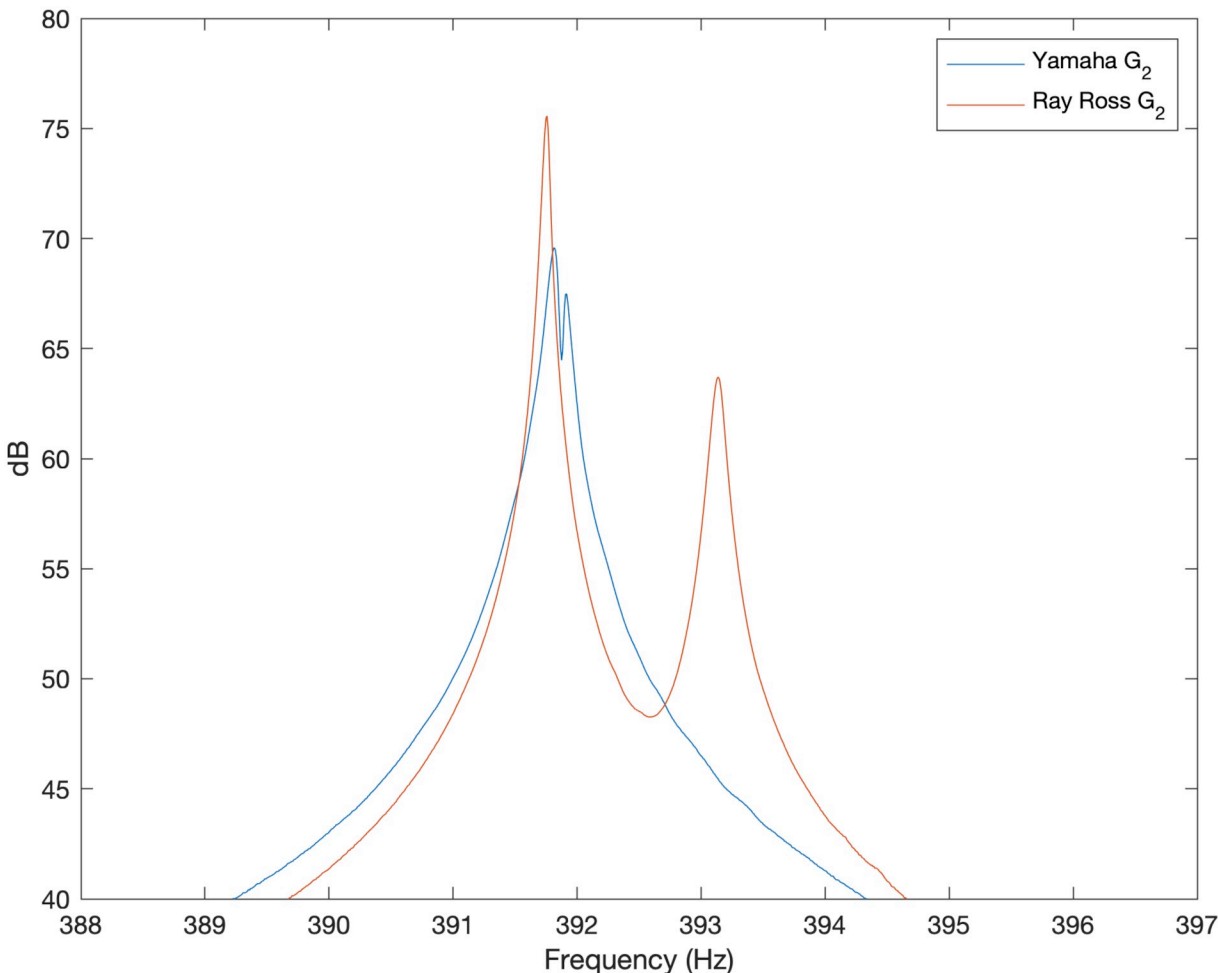

**Fig 7. Peak splitting example.** Experimentally measured sound spectrum around the fourth resonant frequency for the stock Yamaha bridge versus the Ray Ross bridge for the note $G_2$ as measured with optical pickups.

end. It can be expected that vibration of the string in the direction perpendicular to the body, on the other hand, is insufficient to overcome static friction (induced by string tension) between the loop of the core and the ball end, and therefore the core is unable to rotate around the ball end when moving within the plane perpendicular to the body. Modes of vibration featuring motion perpendicular to be the body therefore have a fixed end at the junction of the core wire (as the loop of core is at a non-negligible angle with respect to the axis of motion and therefore cannot be forced into significant length changes by small amplitude/linear motion of the string). Measuring the distance between where the string leaves contact with the ball end and junction of the loop gives a length difference of around 3 ± 1 mm. Comparing this length difference to the nominal string length of 864 mm gives an expected difference in pitch of approximately $\mathbb{I} = 1200 \log_2 \left( 864/(864 - 3) \right) \approx 6 \pm 2$ cents between the motion in the plane of and perpendicular to the body. This is within error bounds for the experimental observation. It may be noted that fretting the string at the 12th fret would approximately half the sounding length and thus approximately double the cents difference between the two peaks.

The resulting inharmonicity experimental data is compared to the theoretical predictions obtained using Eq 4 in Fig 8. It is clear that the perturbation method including a

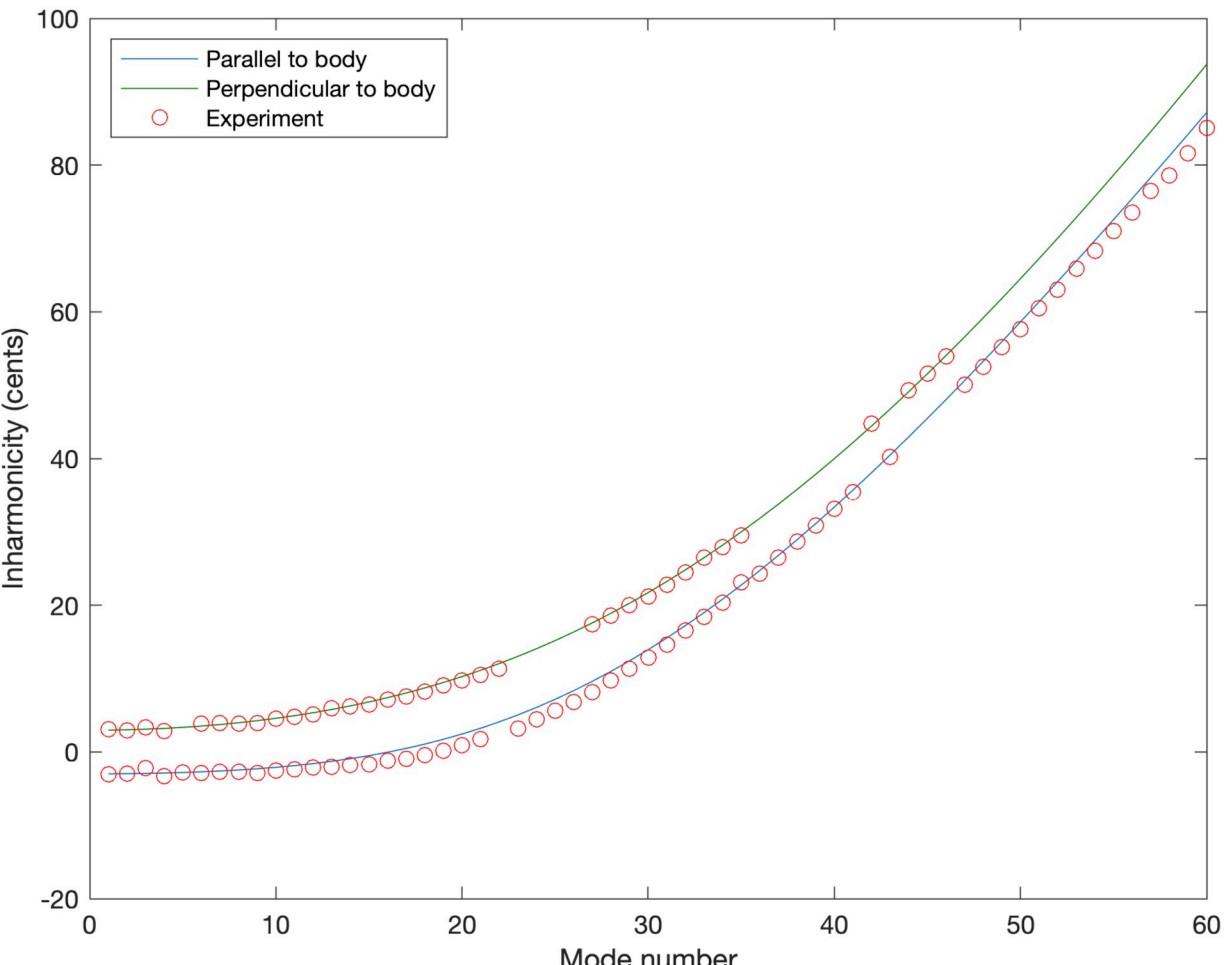

**Fig 8. Inharmonicity.** Experimentally determined inharmonicity for the Ray Ross bridge for the note $G_2$ as measured with optical pickups versus perturbation theory from Eq 4 for motion parallel to and perpendicular to the body.

fictitious winding to model the effect of the loop/knot is successful in predicting the inharmonicity.

Returning to theoretical predictions, in Eq 8 it was predicted that the inharmonicity reduction is most effective when the length of the mass perturbation is an eighth of the wavelength and this should occur around the 11th or 12th modes for vibration parallel to the body and around the 14th mode for vibration perpendicular to the body (given the total knot length of 19 mm and total string length of 864 mm). As seen in Figs 6 and 8, the inharmonicity is strongly reduced by the mass in the knot below these mode numbers and increases at a similar rate to an unperturbed string when going to higher mode numbers (while still starting from a lower baseline inharmonicity) hence giving experimental verification of this effect.

As noted above, the optical pickup is centred around 34mm from the start of the sounding length leading to failures in peak detection around the 25th mode (and integer multiples thereof) due to nodes of vibration near the pickup position for the Yamaha stock bridge. For the Ray Ross bridge, the mass increase in the knot shortens the wavelength there and moves the node positions slightly towards the start of the sounding length meaning the motion parallel to the body shows peak detection failure at mode 22 (and in a region around double that

number). This would be consistent with the end of an unperturbed string lying approximately 5 mm behind the actual start of the sounding length of the real string, and this is therefore consistent with the inharmonicity reduction of approximately $1200 \log_2 ((864 + 5)/864) = 10$ cents seen in the experiments. The vibrations perpendicular to the bridge have detection failures centred around the 24th mode and this is consistent with the optical pickup being 34—3 = 31 mm from the start of the sounding length since the sounding length starts at the junction of the loop (with the mass increase in the knot moving the node positions to a similar extent).

## Peak splitting in musical context

Peak splitting is clearly a significant feature of the sound when the loop/knot is included in the sounding length of the string. In order to give context for the peak splitting observed in the Ray Ross bridge, it is useful to note that peak splitting is common in musical context and to review some examples.

### Grand piano

All but the lowest notes on the grand piano have two or three strings struck simultaneously by each key. This topic is discussed in detail in Benade [11]. If the multiple strings of a single note were sounding perfectly in unison then the strings are all pushing the soundboard in phase, resulting in rapid radiation of energy away from the string, giving a reduced sustain and uninteresting, flat sound. It has been noted that listeners prefer a tuning difference of around 1 cents or 2 cents [11]. As an example, the note $G_2$ was played on a Yamaha C3X (6'1") grand piano, tuned by Iain Ovenstone at the Laidlaw Music Centre at the University of St Andrews and recorded using a Rode NT5 condenser microphone into an Allen and Heath DT168 Dante Audio Expander. The resulting spectrum, calculated in MATLAB, is shown, zoomed in around the 4th resonance, in Fig 9. This features two peaks separated by around 3 cents due to the two strings used for the $G_2$ note being deliberately tuned to have different vibrational frequencies in order to achieve the lively, pulsating sound and increased sustain desired from a grand piano.

### Bass guitar through an electronic chorus effect

The note $G_2$ was played on the open G string of a Sadowsky MV5 bass guitar (featuring magnetic pickups) with the pickup blend knob set to give the signal from the neck pickup. Full volume was selected and the push pull pot was pulled out to deactivate the onboard preamp/tone control circuit. This model of bass features a bridge with conventional, Fender inspired, saddles. The signal from a full pluck was measured using a RME Fireface 400 both direct from the bass into an "instrument" jack input (to give the "clean" signal) and with a Boss Chorus CE-3 chorus pedal inserted into the signal chain. Mono D+E mode was selected so that the (mono) signal coming from the pedal consisted of a the direct signal from the pass plus a signal with vibrato (frequency modulation) added, as is standard for a mono chorus effect. The rate control on the chorus pedal (which sets the modulation frequency) was set at halfway (12 o'clock) and the depth control (which sets the peak frequency deviation) was set approximately a third of the way up (half past 10 o'clock). The resulting spectra, produced using MATLAB and zoomed in around the 4th resonance, are shown in Fig 10.

The clean signal (no chorus) from this bass features split peaks that are separated by around 1 cent (which is greater than the 0.4 cents difference observed in Fig 7). This level of (mild) peak splitting can be attributed to the effect of the non-uniform magnetic field around the pickups reducing the vibration frequency slightly for motion in the direction perpendicular to the body [10]. Such double peaks are observed in the output from the magnetic pickup in spite

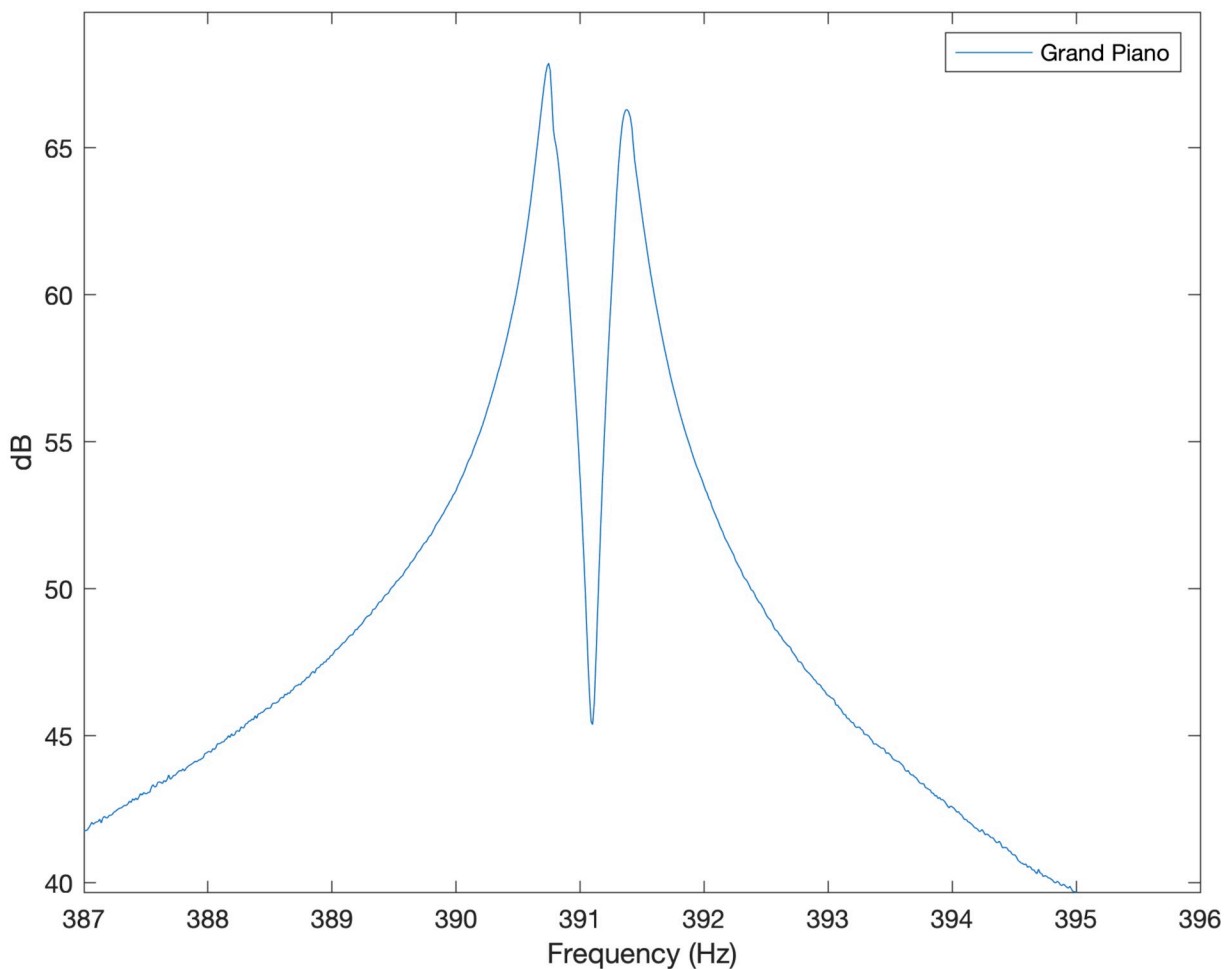

**Fig 9. Grand piano.** Experimentally measured sound spectrum around the fourth resonant frequency for Yamaha C3X grand piano.

of magnetic pickups being more sensitive to vibrations perpendicular to the plane of the body [12]. In the case of the bass with chorus effect, the peak is split into several side bands separated by approximately 1.9 Hz, which corresponds to around 8 cents. This means the chorus pedal was applying a modulation frequency of around 1.9 Hz within the effect circuitry when the "rate" control was in the centre of its range. The results show some similarity to the spectrum produced by the Ray Ross bridge in Fig 7 in terms of peak spacing, although the chorus has more than two peaks present for each mode number.

### Experimental comparison of bridge designs sustain for all strings

The analysis above has been applied to the thinnest string on a standard bass guitar (the open $G_2$ string). Reduction in inharmonicity for the thicker strings is going to be present but more subtle (with inharmonicity reduction being halved for typical thickest strings as set out in the theory section), since the knot will be a smaller proportion of the overall string mass in comparison to the thinnest string. Peak splitting is going to be the same number of cents for the thicker strings as the loop on the knot is the same fraction of the sounding length. Sustain, on the other hand, is a factor that may vary from string to string with the thicker strings

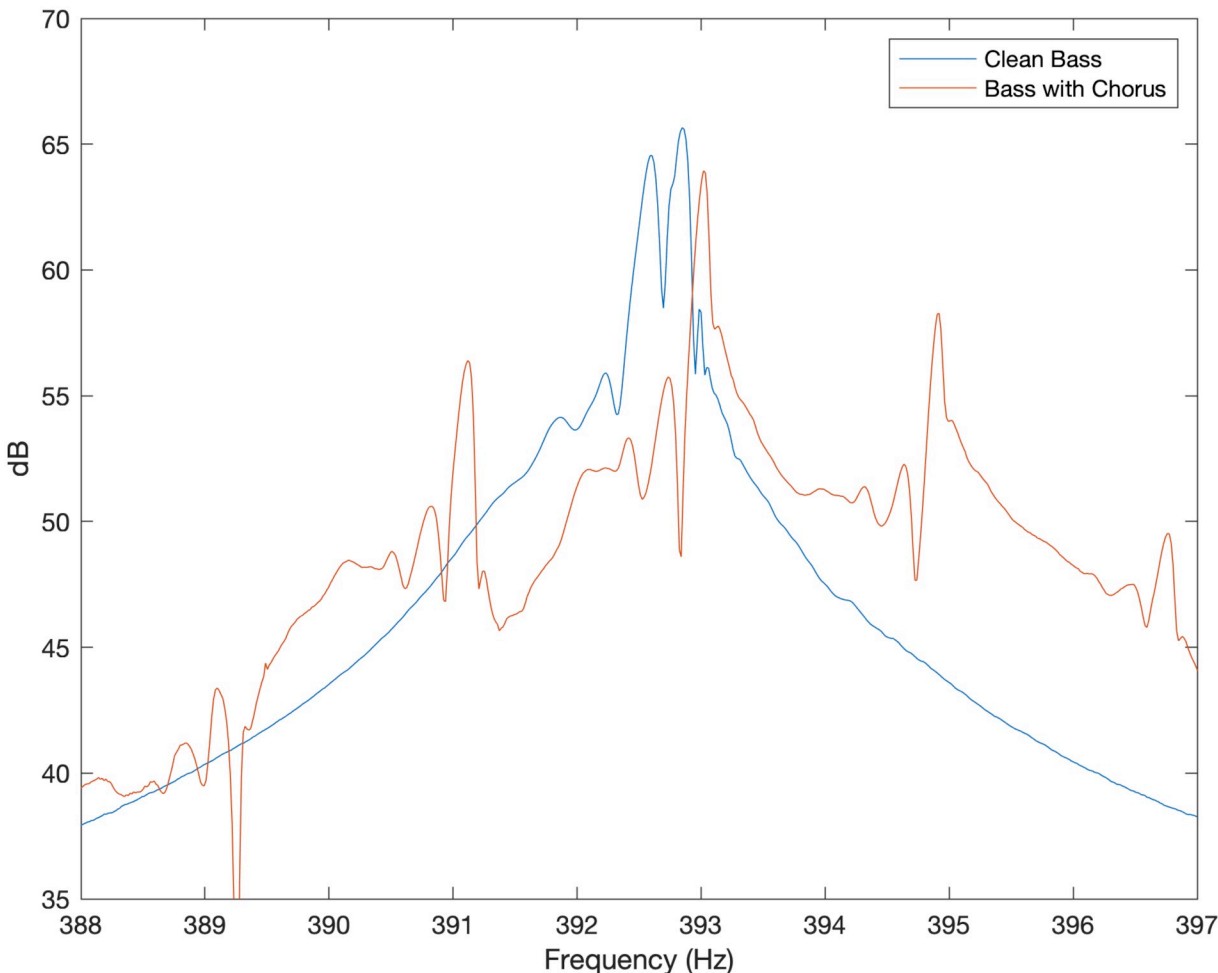

**Fig 10. Bass guitar with and without chorus.** Experimentally measured sound spectrum around the fourth resonant frequency of the note $G_2$ for Sadowsky MV5 bass guitar (featuring magnetic pickups) with and without chorus effect.

potentially having windings crushed into each other where the strings bend over the saddle. Theory describing this process is not currently known to the author so experiments are presented here to investigate.

A set of D'Addario EXL170 nickel wound bass guitar strings were used with the Yamaha BB 350 bass guitar with magnetic pickups installed. Different brands of (roundwound) strings are not expected to give dramatically different results, particularly in terms of relative sustain levels for different bridge designs so using a different brand in comparison to the previous experiment will help establish this. The body of the bass was clamped to a desk using a vice and a Shubb capo was placed behind the first fret for all the following measurements (with the edge of the rubber of the capo in a line with the bottom edge of the first fret where it touches the fretboard) to eliminate the effect of vibrations behind the nut and variations in the break angle at the nut when removing and reinstalling strings. Signals from both stock Yamaha neck and bridge single coil pickups were passed (in parallel) through a jack lead to the instrument input of a FireFace UFX with gain set on +42 dB. Plucking of the string was performed by the author using an orange Tortex plectrum acting on the string above the neck pickup with the peak signal levels recorded varying in the range between -5.4 dB and

-8.5 dB with respect to the clipping level. Each recording was at least 2 minutes and 16 seconds in duration.

Three plucks of each string were recorded with the strings installed on the Ray Ross bridge (labelled "before" below), then five plucks were recorded with the same strings reinstalled with the stock Yamaha bridge and then two measurements were recorded with the same strings reinstalled on the Ray Ross bridge (labelled "after" below). A calculation of the reverberation time in octave bands for each audio measurement was then made using the t60_impulse function with 't30' input argument from the python-acoustics module (acoustics version 0.2.4).

The results are shown in Fig 11 with the average $T_{60}$ of the plucks for each string for each bridge design shown as the data points, with the error bars showing the full range from maximum and minimum of those $T_{60}$ times. As expected, the $T_{60}$ time for the open G string was very similar between the different bridge designs. The errors bars were usually very small as once the string is in position the $T_{60}$ time is quite consistent for plucking at slightly different dynamic levels. There are no examples in the data set where the stock Yamaha bridge data points exceed all the Ray Ross data points in any given frequency band. On the other hand, there are various examples of the sustain being improved on the Ray Ross bridge, particularly

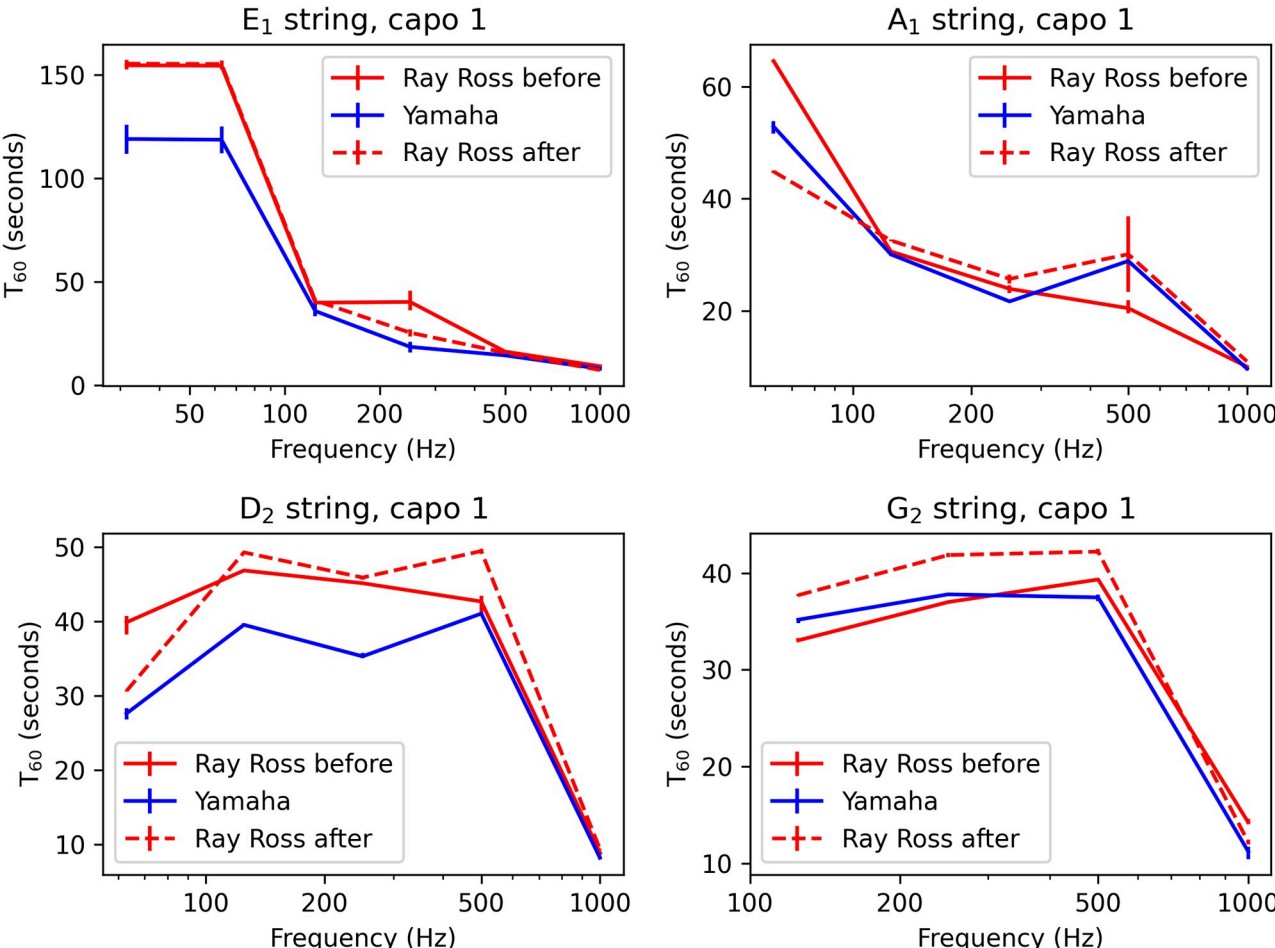

**Fig 11. Sustain for all strings.** Experimentally determined $T_{60}$ time for successive measurements of the same strings on the Ray Ross and the stock Yamaha bridge for the first fret of the $E_1$, $A_2$, $D_2$, and $G_2$ strings, all measured with magnetic pickups. Reverberation times are based on the time taken for the overall envelope of sound to decay within octave bands.

on the lowest octave bands on the $E_1$ and $D_2$ strings. The $A_1$ string also shows signs of the $T_{60}$ time being improved for the Ray Ross design in the "before" measurement but the sustain in the bottom octave band was then lower in the "after" measurement, and this may be due to the kink introduced into the string by installing over the stock Yamaha bridge (and this kink became part of the sounding length in the "after" measurement).

## Conclusions

The acoustical behavior of the Ray Ross bridge has been measured. Sustain is largely unchanged by the inclusion of the knot within the sounding length in comparison to conventional Fender inspired bridge designs for the highest string on the standard bass guitar. There is evidence for the Ray Ross bridge having higher sustain than conventional Fender inspired stock Yamaha bridge design for the thicker strings, with the lowest octave band of the lowest ($E_1$) string showing around a 30% increase in $T_{60}$ sustain level for the low E string on the Ray Ross bridge in comparison to the stock Yamaha bridge across all measurements. This may a result of the windings being forced into one another when the string bends over the saddle in conventional bridge designs. Further research would be useful on this topic.

Each resonance on the string for the Ray Ross bridge design is split into two resonances, parallel to and perpendicular to the plane of the body, due to the length of the loop of core wire only being part of the vibrating length of the string for vibration parallel to the body. This peak splitting is observed to be of the order of ten times greater than the peak splitting due to the magnetic pull of typical bass guitar pickups and is comparable to the pulsations of timbre observed in the deliberate difference in tuning for multiple strings on single piano notes or through the use of an electronic chorus effect. Such an effect is audible for the bass guitar with the Ray Ross bridge fitted, including when the instrument is amplified using magnetic pickups that are most sensitive to motion perpendicular to the plane of the body.

The inclusion of the knot in the sounding length has been shown to reduce inharmonicity and this effect is strongest for modes where the knot takes up an eighth of a wavelength of the mode in question, as predicted by perturbation theory. It should be noted that the reduction of inharmonicity due to the presence of the knot in the sounding length is greatest for the $G_2$ string in a standard bass (where $G_2$ string is the highest pitch sounding) and lower pitched strings will the inharmonicity reduced to a lesser extent due to the proportion of mass in the winding increasing for these strings (lessening the fractional increase in mass in the region of the knot). This means that the inharmonicity is reduced to make the mean of the pairs of resonance closer to a true harmonic series for all strings in the standard bass guitar range ($G_2$ and below), but for the $G_2$ string to the greatest extent.

## Acknowledgments

The author would like to acknowledge Aaron Ray Ross for providing a Ray Ross bridge for testing and would like to thank Dr Charlotte Desvages for helpful discussions on losses in strings and on the use of optical pickups. The authors received no specific funding for this work.

## Author Contributions

**Conceptualization:** Jonathan A. Kemp.

**Data curation:** Jonathan A. Kemp.

**Formal analysis:** Jonathan A. Kemp.

**Investigation:** Jonathan A. Kemp.

**Methodology:** Jonathan A. Kemp.

**Software:** Jonathan A. Kemp.

**Writing – original draft:** Jonathan A. Kemp.

**Writing – review & editing:** Jonathan A. Kemp.

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
