## [Decision Letter · Decision Letter 0]

2 Aug 2023

PONE-D-23-19708The acoustical behavior of a bass guitar bridge with no saddlesPLOS ONE

Dear Dr. Kemp,

Thank you for submitting your manuscript to PLOS ONE. After careful consideration, we feel that it has merit but does not fully meet PLOS ONE’s publication criteria as it currently stands. Therefore, we invite you to submit a revised version of the manuscript that addresses the points raised during the review process.

We look forward to receiving your revised manuscript.

Kind regards,

Khalil Abdelrazek Khalil, Ph.D.

Academic Editor

PLOS ONE

Journal Requirements:

"No funding was applied. All items used in the study were purchased by the author, or provided free in the case of the Ray Ross bridge, or was equipment owned by the University of St Andrews."

Reviewers' comments:

Reviewer's Responses to Questions

**Comments to the Author**

1. Is the manuscript technically sound, and do the data support the conclusions?

Reviewer #1: Yes

Reviewer #2: Partly

2. Has the statistical analysis been performed appropriately and rigorously? 

Reviewer #1: N/A

Reviewer #2: No

3. Have the authors made all data underlying the findings in their manuscript fully available?

Reviewer #1: No

Reviewer #2: Yes

4. Is the manuscript presented in an intelligible fashion and written in standard English?

Reviewer #1: Yes

Reviewer #2: Yes

5. Review Comments to the Author

Reviewer #1: This is a clear account of a well-defined question, and I only have very minor comments.

First, the author has surely not complied with the instructions about data availability? You specifically forbid "Available from the author"

Throughout, we have an odd mixture of inches and mm as units. I understand that strings may be marketed in inches, but for a scientific publication it would surely be better to use SIU throughout?

In the discussion on the Grand Piano, it seems very odd not to reference the famous paper by Weinreich.

Finally, a few typos:

Line 55 "with first with"

Line 211 "sting"

Line 239 "inharomonicty"

Line 321 "deigns"

Reviewer #2: Review: PONE-D-23-19708

Title: The acoustical behavior of a bass guitar bridge with no saddles

The paper deals with the comparative acoustic analysis of bass guitar strings depending on the type of bridge used (conventional, respectively Ray Ross bridge). The article is interesting both theoretically and experimentally. However, there are some aspects that should be improved so that it is understood.

The first aspect that I recommend is related to the organization of the chapters, as there should be a section in which the two methods addressed (experimental and analytical modelling) are presented, perhaps even the author will introduce a logical scheme of the research concept presented in this article.

Thus, the methods used in this study are: analytical, experimental and data processing.

The experimental ones aim at three components: testing the G string with two types of bridges, testing in a musical context (comparisons between all strings) and finally, testing the piano string, respectively electronic chorus effect.

The theoretical chapter, Theory for the effect of the knot, could be moved higher, after the Introduction section.

Introduction

In the explanations given in the Introduction, paragraphs 19 - 28, a figure should be added to exemplify the two ways of holding the strings on the bridge.

Experimental comparison of bridge designs for G string using optical pickups

In the description of the experiment, can you indicate at what distance from the bridge the optical pickup was placed?

Why did you choose to analyse the G string and not the others?

In the section Experimental comparison of bridge designs sustain for all strings, you mentioned that the comparative acoustic analysis of the 4 strings was done with D'Addario EXL170 strings, unlike the first stage of the experiment in which the G string was made by Sadowsky Blue. Why didn't you use the same type of strings throughout your studies? What implications does the type of string have on inharmonicity and sustain?

Both in the section and Experimental comparison of bridge designs for G string using optical pickups and In the section Experimental comparison of bridge designs sustain for all strings, the graphs with the time analysis of the damped signal of the tested strings should be added.

Finally, I recommend the major revision of the paper so that it can be published.

6. PLOS authors have the option to publish the peer review history of their article (what does this mean?). If published, this will include your full peer review and any attached files.

Reviewer #1: No

Reviewer #2: No

---

## [Author Response · Author response to Decision Letter 0]

21 Aug 2023

Thank you for the helpful comments and please do review the detailed responses in the Response to Reviewers.pdf and edited manuscript etc. Yours, Jonathan Kemp

---

## [Decision Letter · Decision Letter 1]

25 Sep 2023

The acoustical behavior of a bass guitar bridge with no saddles

PONE-D-23-19708R1

Dear Dr. Kemp,

We’re pleased to inform you that your manuscript has been judged scientifically suitable for publication and will be formally accepted for publication once it meets all outstanding technical requirements.

Kind regards,

Khalil Abdelrazek Khalil, Ph.D.

Academic Editor

PLOS ONE

Additional Editor Comments (optional):

Reviewers' comments:

Reviewer's Responses to Questions

**Comments to the Author**

1. If the authors have adequately addressed your comments raised in a previous round of review and you feel that this manuscript is now acceptable for publication, you may indicate that here to bypass the “Comments to the Author” section, enter your conflict of interest statement in the “Confidential to Editor” section, and submit your "Accept" recommendation.

Reviewer #1: All comments have been addressed

Reviewer #2: All comments have been addressed

2. Is the manuscript technically sound, and do the data support the conclusions?

Reviewer #1: (No Response)

Reviewer #2: Yes

3. Has the statistical analysis been performed appropriately and rigorously? 

Reviewer #1: (No Response)

Reviewer #2: N/A

4. Have the authors made all data underlying the findings in their manuscript fully available?

Reviewer #1: (No Response)

Reviewer #2: Yes

5. Is the manuscript presented in an intelligible fashion and written in standard English?

Reviewer #1: (No Response)

Reviewer #2: Yes

6. Review Comments to the Author

Reviewer #1: (No Response)

Reviewer #2: The author took into account all the recommendations and improved the manuscript accordingly. The manuscript may be accepted for publication.

7. PLOS authors have the option to publish the peer review history of their article (what does this mean?). If published, this will include your full peer review and any attached files.

Reviewer #1: No

Reviewer #2: No

---

## [Editor Report · Acceptance letter]

2 Oct 2023

PONE-D-23-19708R1 

The acoustical behavior of a bass guitar bridge with no saddles 

Dear Dr. Kemp:

I'm pleased to inform you that your manuscript has been deemed suitable for publication in PLOS ONE. Congratulations! Your manuscript is now with our production department. 

Kind regards, 

on behalf of

Dr. Khalil Abdelrazek Khalil 

Academic Editor

PLOS ONE